# The Effectiveness of Using Autologous Fat in Temporomandibular Joint Ankylosis Treatment with Interposition Arthroplasty Method: A Systematic Literature Review

**DOI:** 10.3390/healthcare13172241

**Published:** 2025-09-08

**Authors:** Gerda Kilinskaite, Nida Kilinskaite, Marijus Leketas

**Affiliations:** 1Department of Maxillofacial Surgery, Lithuania University of Health Sciences, LT-47181 Kaunas, Lithuania; marijus.leketas@lsmu.lt; 2Faculty of Odontology, Lithuanian University of Health Sciences, LT-47181 Kaunas, Lithuania; nida.kilin@gmail.com

**Keywords:** temporomandibular joint, ankylosis, autologous fat, interpositional arthroplasty, adipose tissue, fat graft

## Abstract

**Relevance of the problem and aim of the work**: Ankylosis of the temporomandibular joint (TMJ) affects physical, psychological, and social well-being and quality of life. One of the most frequently used surgical interventions for the treatment of temporomandibular joint ankylosis is interpositional arthroplasty, particularly in cases where joint preservation is feasible, with different autologous fats: dermis fat, buccal fat pad, and full thickness skin-subcutaneous fat. The aim of the work was to evaluate the efficiency of using different autologous fats in temporomandibular joint ankylosis treatment with interposition arthroplasty method. **Materials and Methods**: This systematic literature review was conducted according to PRISMA guidelines and registered in the PROSPERO database (CRD420251038325). A comprehensive search was performed in PubMed, the Cochrane Library, and ScienceDirect databases using combinations of keywords: (temporomandibular joint disorders OR temporomandibular joint) AND (adipose tissue or autologous) AND (ankylosis OR arthroplasty). Inclusion criteria were clinical studies conducted on human subjects, written in English, that evaluated the use of autologous fat in interpositional arthroplasty for TMJ ankylosis. The main outcome measures included postoperative maximum mouth opening (MMO), pain intensity, and relative fat volume contraction. Risk of bias was assessed using the Cochrane RoB 2 tool for randomized controlled trials and the Newcastle–Ottawa Scale for cohort studies. Most included studies were of moderate to high quality. **Results**: A total of 20 publications were selected, including a total of 369 patients. In a qualitative analysis, the best results for maximal opening of mouth (MOM) at 3, 6, 12, and more than 12 months were obtained with dermal fat. After 3 months, the MOM was 40.0 ± 2.7 mm, after 6 months—40.80 ± 4.26 mm, after 12 months—41.9 ± 4.0 mm, after more than 12 months—43.5 mm. The lowest pain intensity was observed using dermal fat taken from the iliac crest region. The rate of volumetric fat shrinkage was greater using buccal fat pad than dermis fat. **Conclusions**: The most commonly used types of autologous fat in interposition arthroplasty in ankylosis are the following: dermal fat from the abdominal region (iliac crest, subumbilical area, groin), buccal fat pad and full-thickness subcutaneous fat. The best results after the surgical treatment of TMJ ankylosis with interposition arthroplasty are obtained using dermis fat.

## 1. Introduction

Ankylosis of the temporomandibular joint (TMJ) is defined as a bony or fibrous adhesion of the anatomical components of the joint, resulting in loss of function. In many cases, fibrous ankylosis progresses to bony ankylosis [1]. The choice of management depends on the type and severity of the ankylosis; while fibrous ankylosis may respond to conservative therapy, bony ankylosis typically requires surgical intervention. This pathology disrupts physiological functions such as speech and mastication, causes changes in facial esthetics, can lead to life-threatening airway obstruction, and complicates oral hygiene [2]. TMJ ankylosis affects physical, psychological, and social well-being, as well as quality of life. The etiology of TMJ ankylosis includes trauma (the most common cause), systemic diseases such as ankylosing spondylitis or rheumatoid arthritis, otitis media, TMJ infections, and complications following surgery or local inflammation [2,3,4,5]. Pathogenesis typically begins with intra-articular bleeding or inflammation, which leads to fibrosis, cartilage degeneration, and ultimately bony or fibrous fusion of the joint surfaces. If untreated, repeated cycles of injury and healing promote progressive ossification and joint immobility.

The primary treatment method for TMJ bony ankylosis is surgical intervention, aimed at restoring TMJ mobility and function. However, the choice of surgical techniques remains controversial. The available surgical options for TMJ ankylosis include the following: gap arthroplasty, interpositional arthroplasty, total joint reconstruction (e.g., alloplastic prosthesis), and condylectomy with distraction osteogenesis or costochondral grafting [6,7]. One of the most frequently used surgical interventions, due to its low recurrence rate, is interpositional arthroplasty using various materials [8]. The most common surgical approach for interpositional arthroplasty is the preauricular incision, often extended or combined with a submandibular or temporal incision depending on the complexity of ankylosis and graft harvesting requirements. The goal of interpositional arthroplasty is to create a gap between the condylar fossa of the temporal bone and the separated mandible, filling it with autogenous or alloplastic grafts to prevent the recurrence of ankylosis [9]. Interpositional materials are inserted into this gap, and the appropriate selection of these materials contributes to achieving good long-term outcomes [10]. The surgical site is irrigated, hemostasis is achieved, and layered closure is performed. Care is taken to minimize tension on the facial nerve branches and skin flaps [10].

Nowadays, increasing attention is being given to the use of autogenous fat in the treatment of TMJ ankylosis by means of interpositional arthroplasty using various materials [8]. These materials include autologous tissues such as temporalis muscle or fascia, costochondral grafts, dermis-fat grafts, and alloplastic substitutes like silicone, acrylic, or titanium implants [9,10]. However, autologous fat grafts—particularly dermal and buccal fat—have gained favor due to their biocompatibility, ease of harvesting, adaptability to irregular surfaces, and ability to prevent re-ankylosis by occupying dead space and limiting hematoma formation [11]. Autologous fat is used during ankylosis management to prevent heterotopic bone formation [11]. The purpose of using autologous fat in interpositional arthroplasty is to fill the space of the articular disk after osteotomy, thereby preventing the separate elements of the TMJ from fusing during the healing process. Additionally, filling the articular disk space with autologous fat eliminates the empty space that could otherwise allow for hematoma formation, which is a contributing factor in the development of reankylosis [3].

The most commonly used autogenous fat grafts are derived from the dermis, buccal fat pads, and full-thickness skin-subcutaneous tissue. The lower abdominal region is typically the donor site for dermal fat grafts due to the relatively simple harvesting technique, ease of manipulation, and good adaptation in the recipient site [6]. The use of buccal fat has increased over the past decade because of its easier local accessibility and adaptability [7]. Full-thickness skin-subcutaneous fat grafts, which include the epidermal layer, provide structural stability to the tissue, helping to preserve the fat and prevent its breakdown, degeneration, and mobility [8].

Despite the increasing use of autologous fat in TMJ ankylosis surgery, the available literature remains fragmented, and a comprehensive synthesis of comparative outcomes among fat graft types is lacking. No prior systematic review has evaluated the clinical effectiveness, complications, and long-term outcomes associated with different types of autologous fat. Therefore, this review is necessary to consolidate evidence, identify best practices, and guide clinicians in selecting optimal interpositional materials for TMJ ankylosis treatment.


**Primary Objective:**


To evaluate the effectiveness of autologous fat grafts in the treatment of temporomandibular joint (TMJ) ankylosis using the interpositional arthroplasty method.


**Secondary Objectives:**
(1)To identify the types of autologous fat used in interpositional arthroplasty for TMJ ankylosis.(2)To compare the effects of different types of autologous fat on maximal mouth opening (MMO) postoperatively.(3)To compare postoperative pain intensity associated with different autologous fat grafts.(4)To evaluate the extent of fat volume contraction among various autologous fat grafts used in TMJ ankylosis treatment.


## 2. Article Selection Criteria, Search Methods, and Strategy

### 2.1. Systematic Review Protocol

This study has been registered in the PROSPERO international systematic review registry. The registration number is CRD420251038325. The protocol for the systematic review of the scientific literature was prepared in advance following the PRISMA [12] guidelines for systematic reviews (Figure 1). The research question for the systematic literature review was formulated using the PICO methodology, considering the pathology, intervention, comparison, and outcomes. The application of the PICO method is presented in a table (see Table 1).

### 2.2. Article Inclusion Criteria

The scientific publications included in this systematic literature review were selected based on well-defined inclusion and exclusion criteria. Studies were eligible for inclusion if they investigated the effectiveness of autologous fat in the treatment of temporomandibular joint (TMJ) ankylosis using the interpositional arthroplasty technique. Only clinical studies conducted on human subjects and published in English were considered, including clinical trials, retrospective analyses, and prospective cohort studies.

Publications were excluded if they were meta-analyses, systematic reviews, case reports, theses, or books. Additional exclusion criteria included studies conducted on animals, in vitro investigations, articles not published in English, those analyzing conservative (non-surgical) treatment methods, or those focusing on the use of autologous fat for TMJ conditions other than ankylosis.

### 2.3. Article Search Methods

The publication search and selection were conducted by two researchers (G. K and N. K.), in consultation with the supervisor (M. L.). The data search was conducted from 22 April 2025 to 30 May 2025 (the last date for database searches was 30 May 2025) in the PubMed, The Cochrane Library, and Science Direct scientific databases and did not include gray literature or manual searches. This may have led to the potential omission of relevant unpublished or non-indexed studies, using a keyword combination developed through a preliminary literature search, providing the most search results. Additionally, keywords and possible synonyms were selected using the MESH (Medical Subject Headings) vocabulary. Furthermore, four Boolean expressions (“OR” and “AND”) were used to ensure a consistent search across all databases. The keyword combination used was the following: (temporomandibular joint disorders OR temporomandibular joint) AND (adipose tissue or autologous) AND (ankylosis OR arthroplasty)).

During the article selection process, only scientific literature with publicly available abstracts was reviewed, and full-text articles were accessible in the LSMU library. In the first stage of article selection, the titles and abstracts of publications were reviewed, and articles potentially meeting the established inclusion criteria were selected, while duplicates and publications not meeting the criteria were excluded. In the second stage of article selection, full-text publications were analyzed, and articles meeting the inclusion criteria were included in the systematic review, while those not meeting the criteria were excluded, with the reasons for exclusion and the number of excluded publications provided.

The agreement between the two reviewers during the study selection process was assessed using Cohen’s kappa coefficient (κ), which demonstrated substantial agreement (κ = 0.79), indicating a high level of internal consistency in article inclusion.

### 2.4. Data Collection

In this systematic review, data from the articles were collected by completing tables, including the most relevant information: main author, year of publication, interpositional material, surgical method, follow-up period, patient age, study design, number of subjects, postoperative maximum mouth opening (MMO) (at 3 months, 6 months, 12 months, and more than 12 months). No statistical software was used, as the data were synthesized qualitatively. Numerical outcomes from individual studies were summarized in tables but not statistically analyzed or pooled due to heterogeneity.

### 2.5. Variables

For the evaluation and comparison of results, the main variables considered were the maximum mouth opening sizes, pain intensity, and relative fat volume contraction as studied in the publications.

### 2.6. Assessment of the Risk of Bias in Studies

The risk of bias in the included scientific publications was assessed using the Cochrane Risk of Bias 2 (RoB 2) tool [13] for randomized controlled trials and the Newcastle–Ottawa Scale [14] for cohort studies.

The risk of bias in randomized controlled trials was determined based on the RoB 2 tool. The level of bias (low, moderate, high) was assessed by focusing on different aspects of study planning, execution, and reporting.

The quality assessment of cohort studies was carried out using the Newcastle–Ottawa scale. This assessment scale includes a “star system,” where the study is analyzed from three broad perspectives: selection of study groups, comparability of groups, and outcome assessment. These criteria were scored, and the sum of the points indicated the methodological quality rating of the publications. Articles that scored 7–9 points were considered of high quality, 6–5 points as moderate quality, and less than 5 points as low quality.

## 3. Data Organization and Analysis

### 3.1. Results of the Data Search

The protocol for the systematic review of scientific literature was developed following the PRISMA [12] recommendations for systematic reviews. Using a combination of keywords in the scientific databases PubMed, The Cochrane Library, and Science Direct, 1330 publications were identified. After removing duplicates, 929 publications remained.

In the first stage of the scientific publication selection process, the titles and abstracts of the remaining articles were reviewed. After evaluating the suitability of the articles and applying the inclusion criteria, 33 publications were selected for full-text analysis. During the full-text analysis, 13 publications were excluded. The excluded articles did not meet the inclusion criteria: the treatment of TMJ ankylosis was conservative, or the articles did not align with the objective of the study—they did not investigate the use of autologous fat in interpositional arthroplasty for the treatment of TMJ ankylosis. In total, 20 publications were included in the qualitative data analysis [15,16,17,18,19,20,21,22,23,24,25,26,27,28,29,30,31,32,33,34]. A detailed data search protocol is presented in a figure (see Figure 1).

Due to significant clinical and methodological heterogeneity among the included studies—such as variation in follow-up durations, graft harvesting sites, surgical protocols, and reporting methods—a meta-analysis was not performed. As a result, statistical heterogeneity (e.g., I^2^ statistic) and publication bias (e.g., funnel plot, Egger’s test) were not quantitatively assessed. The synthesis of the results was conducted using a qualitative approach.

Because no quantitative pooling of data was performed, statistical models such as fixed-effect or random-effect were not applied.

### 3.2. Assessment of the Risk of Bias in Studies

The risk of bias in the selected scientific publications was assessed using the following: for randomized clinical trials—the Cochrane RoB2 assessment tool [13], and for cohort-type studies—the Newcastle–Ottawa Scale [14].

After assessing 4 randomized clinical trials with the Cochrane RoB2 tool, 3 publications were classified as having a low risk of bias [16,17,23], and 1 study was classified as having a moderate risk of bias [24]. The assessment of bias using the RoB2 tool is presented in Table 2.

**Table 2 healthcare-13-02241-t002:** Risk assessment of articles using the Cochrane (RoB 2) tool.

Articles	Random Sequence	Deviations from the Planned Intervention	Missing Outcome Data	Outcome Assessment	Selective Reporting of Results	Overall Assessment
D. Mehrotra et al., 2008, [16]	+	+	+	+	+	+
N.N. Andrade et al., 2020, [17]	+	+	+	+	+	+
A. Roychoudhury et al., 2020, [23]	+	+	+	+	+	+
A.Roychoudhury et al., 2017, [24]	-	-	-	+	-	-

“+”—low risk of bias; “-”—moderate risk of bias.

In assessing the risk of bias in cohort-type studies using the Newcastle–Ottawa Scale, the majority of studies were classified as high quality [15,18,19,20,21,22,25,26,28,29,30,33,34], but 3 studies were categorized as moderate quality [27,31,32]. Possible errors may have occurred because some data were not fully described in the publications, which could have impacted the results. A detailed analysis of the risk of bias is presented in Table 3.

**Table 3 healthcare-13-02241-t003:** Risk assessment of articles using the Newcastle–Ottawa assessment scale.

Articles (Main Author and Year)	Selection				Comparability (★★)	Outcomes			Study Assessment
	1 * (★)	2 * (★)	3 * (★)	4 * (★)		5 * (★)	6 * (★)	7 * (★)	
J. Yazdani et al., 2010, [15]	★	★	★	★	★	★		★	7
S.A. Rahman et al., 2020, [18]	★	★	★	★	★	★	★		7
M. Karames e et al., 2013, [19]	★	★	★	★	★	★	★	★	8
G. Dimitrou lis et al., 2004, [20]	★	★	★	★	★★	★	★	★	9
M. Younis et al., 2021, [21]	★	★	★	★	★	★	★	★	8
A.F. Hegab et al., 2015, [22]	★	★	★	★	★	★	★	★	8
V. Bansa l et al., 2015, [25]	★	★	★	★	★★	★	★	★	9
S. Gaba et al., 2012, [26]	★	★	★	★	★	★	★		7
A.A. Ibikunle et al., 2018, [27]	★	★		★	★		★	★	6
R.F. Elgazzar et al., 2010, [28]	★	★	★	★	★	★	★	★	8
V.L. Malhotra et al., 2019, [29]	★	★	★	★	★★	★	★	★	9
V. Singh et al., 2011, [30]	★	★	★	★	★	★	★	★	8
V. K. Sharma et al., 2020, [31]	★	★	★	★	★★				6
V.K. Sharma et al., 2019, [32]	★	★	★	★	★			★	6
Y. Ma et al., 2019, [33]	★	★	★	★	★★	★	★	★	9
A.Thang avelu et al., 2011, [34]	★	★	★	★	★	★	★	★	8

* Explanation. Selection criteria: 1—representativeness of the study, 2—selectivity of the subjects, 3—determination of the intervention, 4—outcome not pre-specified at the beginning of the study; outcome criteria: 5—outcome assessment, 6—sufficient follow-up period, 7—adequacy of follow-up. The stars (★) represent how well a study meets specific quality criteria. 7–9 stars → High-quality study. 4–6 stars → Moderate quality. 0–3 stars → Low quality.

### 3.3. Characteristics of Included Studies

A total of 20 publications [15,16,17,18,19,20,21,22,23,24,25,26,27,28,29,30,31,32,33,34] were included in the systematic literature review. The included articles described randomized clinical trials as well as controlled retrospective and prospective studies involving human subjects. In total, 369 patients were studied, with sample sizes ranging from 4 to 36 participants. The studies assessed the effects of various types of autologous fat on temporomandibular joint (TMJ) function in the treatment of TMJ ankylosis using interpositional arthroplasty. Ten studies analyzed dermal fat [15,16,17,18,19,20,21,22,23,24], eleven studies evaluated buccal fat [23,24,25,26,27,28,29,30,31,32,33], and one study assessed the effect of full-thickness skin-subcutaneous fat on treatment outcomes [34]. The most relevant data were extracted and organized based on the following: lead author, year of publication, surgical technique, follow-up period, age of participants, study design, number of participants, and changes in maximum mouth opening (MMO) at different time intervals. The data are presented in Table 4. Due to heterogeneity among the included studies, a meta-analysis was not performed.

**Table 4 healthcare-13-02241-t004:** Characteristics of the articles.

Authors	Material	Surgical Method	Follow-Up Period	Patients’ Age (Years)	Study Design	Number of Subjects	Before Surgery, MMO (mm)	3 Months After Surgery, MMO (mm)	6 Months After Surgery, MMO (mm)	12 Months After Surgery, MMO (mm)	More than 12 Months After Surgery, MMO (mm)
Randomized Clinical Trials											
D. Meh rotra et al., 2008 [16]	Dermal fat (iliac crest region)	IA	24 ± 7 months	6.5 (4–15)	Randomized clinical trial	8	3.9 ± 3.2	-	33.3 ± 4.3	-	
	Temporalis muscle flap		26 ± 5 months			9		-	25.9 ± 5.4	-	
N.N. A ndrade et al., 2020 [17]	Dermal fat (lower abdominal area)	IA	3 years	24.90 ± 12.55	Randomized clinical trial	22	4.0 ± 1.56	-	40.80 ± 4.26	41.80 ± 2.82	41.20 ± 4.69 (after 2 years)
	Gap arthroplasty		3 years				3.60 ± 1.78		38.20 ± 2.53	38.80 ± 2.30	39.50 ± 2.46 (after 2 years)
A. Roycho udhury et al., 2020 [23]	Buccal fat	IA	12 months	12.6 + 8.5	Randomized clinical trial	18	6.8 + 4.4	-	-	30.6 + 6.3	-
	Dermal fat (lower abdominal area)			14.3 ± 8.2			4.2 + 1.2			41.9 + 4.0	
A.Royc houdhu ry et al., 2017 [24]	Buccal fat	IA	1 year	-	Randomized clinical trial	18	-	-	-	30–35	-
	Dermal fat (lower abdominal area)									>40	
Cohort studies											
J. Yazdan i et al., 2010 [15]	Dermal fat (abdominal region)	IA	3 months 3 months	27.8 ± 8.8	Prospective cohort study	10	4.1 ± 1.9	40.0 ± 2.7	-	-	-
	Temporalis muscle flap			25 ± 9.1		10	6.4 ± 3.1	41.6 ± 2.0			
M. Karame se et al., 2013 [19]	Dermal fat (abdominal or inguinal area) + temporal muscle flap	IA	42.09 ± 8.21 months	30 (5–51)	Prospective cohort study	11	5.6 ± 1.22	-	-	-	29.54 ± 3.17 (after 42.09 ± 8.21 months)
G. Dimitro ulis et al., 2004 [20]	Dermal fat (inguinal area) + CCG	IA	41.45 ± 17.7 months	32.5 (18–55)	Retrospective cohort study	11	15.63 ± 3.39	-	-	-	35.72 ± 2.11 (after 2–6 years)
M. Younis et al., 2021 [21]	Dermal fat (inguinal area)	IA	6 months	6.5	Prospective cohort study	15	8.46	-	39.93	-	-
	Temporalis muscle and fascia					15	9.67		33.67		
A.F. Hegab et al., 2015 [22]	Dermal fat (lower abdominal area)	IA	32.5 (24–48) months	18.5 (12–38)	Prospective cohort study	14	2.071	-	-	-	43.5 (24–48 months)
V. Bans al et al., 2015 [25]	Cheek fat + CCG and others	IA	31 (24–36) months	11 (5–17)	Prospective cohort study	11 8	4.9 ± 3.7	-	-	-	32.5 ± 5.0 (after 31 (24–36 months)
S. Gaba et al., 2012 [26]	Cheek fat	IA	6.12 months	-	Prospective cohort study	18	5.11 ± 6.51	-	33.16 ± 5.24	33 ± 5.28	-
A.A. Ibikunl e et al., 2018 [27]	Cheek fat	IA	12 months	13.25 (4–24)	Prospective cohort study	4	3.5 ± 2.9	-	≥35	-	-
R.F. Elgazza r et al., 2010 [28]	Cheek fat + CCG	RA	1 year	19.43 (2–41)	Retrospective cohort study	9	5.3 ± 3.7	-	-	34.3 ± 1.9	-
V.L. Malhotr a et al., 2019 [29]	Cheek fat	Lateral arthroplasty	Up to 2 years 7 months	11.8 ± 3.02	Retrospective cohort study	10	5 ± 4.85	-	-	-	34.7 ± 2.49 (after up to 2.7 years)
V. Singh et al., 2011 [30]	Cheek fat	IA	14.8 (8–22) months	18.1 (8–35)	Retrospective cohort study	10	2.8 ± 1.5	-	-	-	35.1 ± 3.04 (after 14.8 months)
V. K. Sharma et al., 2020 [31]	Cheek fat	IA	3 months	8.86 (8–10)	Retrospective cohort study	18	4.89	28.16	-	-	-
V.K. Sharma et al., 2019 [32]	Cheek fat	IA	3 months	21.7 ± 8.13	Prospective cohort study	25	6.16	33.76	-	-	-
Y. Ma et al., 2019 [33]	Cheek fat + temporal muscle flap	Arthroplasty + distraction osteogenesis	29.6 (16–45) months	7 (4–12)	Prospective cohort study	17	1.4 (0–5)	-	-	-	35.7 (after 29.6 months)
A.Than gavelu et al., 2011 [34]	Full-thickness skin—subcutaneous fat	IA	12–24 months	27.2 (14–39)	Prospective cohort study	7	0–8	-	-	-	34.5 mm (after 12–24 months)

### 3.4. Overview of the Results Presented in the Studies

#### 3.4.1. Key Findings, Their Significance, and Interpretations

All included studies were conducted on human subjects. The influence of autologous fat on the function of the temporomandibular joint (TMJ) was evaluated clinically by measuring the maximum mouth opening (MMO) at various postoperative time points. A qualitative analysis of the studies was performed; quantitative data were not compared due to clinical and methodological heterogeneity among the included publications.

This systematic review included 20 publications [15,16,17,18,19,20,21,22,23,24,25,26,27,28,29,30,31,32,33,34]. Four of these were randomized controlled trials, six were retrospective cohort studies, and ten were prospective cohort studies. The age of participants ranged from 6.5 to 32.5 years, and sample sizes varied from 4 to 36 subjects. The follow-up period for these studies ranged from 3 to 42.09 ± 8.21 months. Among the 20 studies analyzed, 10 used dermal fat (from regions such as iliac crest, lower abdomen, inguinal area), 11 used buccal fat pad, and 1 used full-thickness skin-subcutaneous fat.

Before surgery, the maximum mouth opening ranged from 0 to 15.63 ± 3.39 mm. Three months postoperatively, the MMO ranged from 28.16 to 40.0 ± 2.7 mm [15,31,32]. After 6 months, the MMO ranged from 30.1 to 40.80 ± 4.26 mm [16,17,18,21,26]. After 12 months, the MMO ranged from 30 to 41.9 ± 4.0 mm [17,23,24,26,27,28]. Beyond 12 months, the MMO ranged from 29.54 ± 3.17 to 43.5 mm [17,19,20,22,25,29,30,33,34].

The best outcomes and the highest maximum mouth opening (MMO) at 3, 6, 12, and more than 12 months postoperatively were achieved using dermal fat harvested from the abdominal area [15,22] and the lower abdominal region [17,23].

The lowest MMO at 3 and 12 months was observed with the use of buccal fat pad, while the lowest MMO at 6 and more than 12 months was seen with dermal fat [18,19,24,31].

Physiotherapy was applied in 16 of the included studies [16,17,18,19,20,21,22,23,25,27,28,29,30,31,32,34]. In twelve studies, physiotherapy was initiated on the 1st or 2nd postoperative day [16,17,21,22,23,25,27,28,29,30,31,32], while in four studies it started on the 6th or 7th day after surgery [18,19,20,34].

Pain intensity, measured using the Visual Analog Scale (VAS), ranged from 1.4 ± 0.5 to 1.8 after 1 week of using dermal fat [16,18], from 0.66 to 2.6 ± 0.52 after 1 month, and from 0.0 to 0.6 ± 0.7 after 6 months [16,17,18]. Pain was only evaluated in studies involving dermal fat [16,17,18]. The lowest pain scores at 1 week, 1 month, and 6 months were recorded when dermal fat harvested from the iliac crest area was used [16,18].

Relative fat volume shrinkage was greater when using buccal fat [66.92–67.50%] compared to dermal fat harvested from the abdomen [41.90–45.29%] [23,24].

#### 3.4.2. Dermal Fat Studies

Ten publications investigated the effectiveness of using dermal fat for the treatment of temporomandibular joint (TMJ) ankylosis via interpositional arthroplasty [15,16,17,18,19,20,21,22,23,24]. In all studies included in this systematic review, the donor site for dermal fat was the abdominal region. Some authors specified more precise abdominal donor areas for harvesting and transplanting dermal fat: two publications indicated the iliac crest region [16,18], two specified the lower abdominal area [17,23], and three mentioned the inguinal region [15,20,21]. The follow-up period for these studies ranged from 3 to 42.09 ± 8.21 months.

The maximum mouth opening before surgery ranged from 2.071 to 15.63 mm across the studies [15,16,17,18,19,20,21,22,23]. Postoperative outcomes showed consistent improvement over time. At 3 months, the MMO using dermal fat reached up to 40.0 mm [15]. This continued to increase at the 6-month mark, ranging from 30.1 to 40.8 mm [16,17,18,21], and by 12 months, values between 40.0 and 41.9 mm were observed [17,19,20,22].

In long-term follow-up beyond 12 months, MMO varied from 29.54 to as high as 43.5 mm, highlighting the sustained effectiveness of dermal fat grafts in maintaining joint mobility.

#### 3.4.3. Buccal Fat Pad Studies

The effectiveness of using buccal fat pad for the treatment of temporomandibular joint (TMJ) ankylosis via interpositional arthroplasty was described in eleven publications [23,24,25,26,27,28,29,30,31,32,33]. The follow-up period for these studies ranged from 3 to 31 months.

In studies evaluating buccal fat pad as an interpositional material, the preoperative maximal mouth opening (MMO) ranged from 1.4 mm to 6.8 ± 4.4 mm, indicating a severe restriction in joint function prior to surgery [23,25,26,27,28,29,30,31,32,33]. Following surgical intervention, a gradual improvement in MMO was observed across timepoints. At 3 months postoperatively, the MMO increased to a range between 28.16 and 33.76 mm [31,32], while at 6 months, it reached 33.16 ± 5.24 mm [26]. This improvement was sustained at 12 months, with values ranging from 30 to 35 mm [23,24,26,27,28]. In the long term, beyond 12 months, MMO measurements further increased slightly, ranging from 32.5 ± 5.0 mm to 35.7 mm [25,29,30,33]. These findings suggest that the use of buccal fat pad contributes to consistent postoperative functional recovery of the temporomandibular joint over time.

#### 3.4.4. Full-Thickness Skin-Subcutaneous Fat Studies

One retrospective cohort study investigated the effectiveness of using full-thickness skin-subcutaneous fat for the treatment of TMJ ankylosis via interpositional arthroplasty [34]. The average age of the participants was 27.2 years, and the study included 7 patients. The study was conducted from 12 to 24 months.

In the study assessing full-thickness skin-subcutaneous fat as the interpositional material, the preoperative maximal mouth opening (MBI) ranged from 0 to 8 mm, reflecting significant ankylosis-related limitation [34]. Following surgical treatment, a marked improvement was observed, with MBI increasing to 34.5 mm at 12 to 24 months postoperatively. This result demonstrates the long-term effectiveness of full-thickness skin-subcutaneous fat in restoring functional mouth opening in patients with TMJ ankylosis.

## 4. Discussion of Results

During the systematic literature review, 20 publications were analyzed [15,16,17,18,19,20,21,22,23,24,25,26,27,28,29,30,31,32,33,34]. All included studies were conducted on humans diagnosed with TMJ ankylosis, who underwent surgical treatment—interpositional arthroplasty with autologous fat. A qualitative data analysis was performed, and quantitative data were not compared due to clinical and methodological heterogeneity among the included studies.

The main goal of TMJ ankylosis treatment is to perform osteotomy of the ankylosed joint in order to create a gap that separates the fused segments and maintain this gap by inserting a graft [17]. Compared to other commonly used interpositional materials such as temporalis muscle flaps, costochondral grafts, or alloplastic implants, autologous fat grafts offer several advantages. These include minimal donor site morbidity, reduced risk of immune reaction, and favorable histological behavior in the joint environment. Dermal fat, in particular, has shown lower volume shrinkage and superior functional outcomes in multiple studies included in this review. Its pseudodisk-like function and cushioning effect may contribute to better long-term joint mobility and patient comfort.

When evaluating the scientific articles investigating the treatment of TMJ ankylosis with interpositional arthroplasty using autologous fat, the best results and the greatest maximum mouth opening (MBI) after 3, 6, 12, and more than 12 months were achieved by inserting dermal fat into the gap created during osteotomy, where it performs a pseudodisk function [15,17,22,23]. The favorable outcomes could be attributed to the fact that dermal fat has good adaptability to various surfaces and resistance to pressure [35]. Analyzing the scientific literature investigating the relative volume contraction of fat, it was found that greater contraction occurs in cheek fat (66.92–67.50%) compared to dermal fat (41.90–45.29%) [23,24]. Several factors could explain the greater volume reduction in cheek fat compared to dermal fat: cheek fat cells are smaller than dermal fat cells, there are fewer stem cells in the cheek area, and the volume of cheek fat is relatively limited during mobilization, which results in greater vulnerability to fragmentation during TMJ function [36]. Dermal fat consists of larger cells and has more stem cells, which possess greater lipogenesis potential [37]. Additionally, some authors claim that the dermal layer in dermal fat helps to maintain the fat intact, preventing fragmentation into smaller pieces, and serves as a carrier for the adipose tissue [20,38].

Undoubtedly, early postoperative physiotherapy has an impact on the successful treatment outcome [17,39]. The goal of postoperative physiotherapy is to prevent the formation of new adhesions, the development of masticatory muscle contractures, and to restore normal muscle function. Some authors suggest waiting 5–7 days before starting physiotherapy to allow postoperative pain and swelling to subside, arguing that early initiation of physiotherapy may provoke bleeding and lead to the formation of large hematomas, which complicate healing and increase the likelihood of ankylosis recurrence. However, in the scientific article analyzed, which achieved the best results after 12 months, it is stated that physiotherapy was started on the first day after surgery, and a drain was inserted into the wound area to reduce postoperative hematoma [22]. The use of a drain to reduce postoperative hematoma is also common in orthopedic surgery (e.g., hip joint arthroplasty) [40,41].

The types of autologous fat selected may influence the intensity of pain. Analyzing the studies assessing pain intensity, it was found that the lowest pain levels after 1 week, 1 month, and 6 months were recorded using dermal fat harvested from the iliac crest region [16,18]. One of the articles included in this literature review compared pain intensity during the postoperative period after interpositional arthroplasty with dermal fat and arthroplasty without interpositional materials. The Visual Analog Scale (VAS) scores were lower in the dermal fat group with a statistically significant difference on the 14th day (*p* = 0.029), but after 3 years, there was no statistically significant difference. It is believed that the fat allows the resected bony surfaces to glide smoothly and functions as a pseudodisk [17]. However, we acknowledge that postoperative pain is influenced by numerous confounding factors such as individual pain tolerance, comorbidities, surgical technique, and rehabilitation protocols. Therefore, while pain was reported in several of the included studies and summarized here for completeness, it should not be considered a primary outcome when evaluating the effectiveness of autologous fat grafts. The main function of fat in this context remains the prevention of re-ankylosis by filling the joint space and acting as a physical barrier.

Regarding donor site selection for dermal fat, several studies included in this review indicated that dermis harvested from the iliac crest or lower abdominal area provided superior clinical outcomes. Specifically, the iliac crest region was associated with the lowest postoperative pain intensity [16,18], while the lower abdominal region yielded the highest maximum mouth opening (MMO) values at follow-up [17,23]. These findings suggest that dermal fat grafts obtained from the iliac crest or lower abdomen may be more favorable for interpositional arthroplasty in TMJ ankylosis due to better volume stability, reduced shrinkage, and lower pain scores. However, no studies directly comparing donor sites were designed specifically to compare anatomical donor sites; thus, further research is warranted to confirm the superiority of any single region.

Although the use of autologous fat grafts has demonstrated promising outcomes in TMJ ankylosis treatment, potential risks must also be considered. Complications associated with fat harvesting include donor site morbidity such as pain, hematoma, infection, scarring, and contour irregularities, particularly when fat is harvested from the abdominal or iliac crest regions. In rare cases, patient dissatisfaction with cosmetic outcomes at the donor site may occur. Additionally, improper placement or inadequate volume of the fat graft may affect joint biomechanics and, in some cases, contribute to malocclusion or joint instability. However, none of the studies included in this review reported severe complications directly related to fat harvesting or its intra-articular use. Further high-quality studies are needed to systematically assess these potential risks [17,23].

The lowest maximum mouth opening (MBI) after 3 and 12 months was achieved using buccal fat, while after 6 months and more than 12 months, dermal fat was used [18,19,24,31]. The poor results when using buccal fat during interpositional arthroplasty could be attributed to the previously mentioned properties of buccal fat: smaller cell size, fewer stem cells, and a higher relative fat volume contraction [37]. Interestingly, when evaluating systemic risk biases using the Cochrane RoB 2 tool, the randomized controlled trial that achieved the lowest MBI after 12 months had a medium systemic risk, raising concerns about the reliability of the study’s methodology [24]. It can be hypothesized that the poor results when using dermal fat could be influenced by the later initiation of physiotherapy. In the prospective cohort studies by S.A. Rahman and co-authors [18] and M. Karamese and co-authors [19], which achieved the lowest MBI after 6 and more than 12 months using dermal fat, physiotherapy was started 7 days after the operation, unlike most of the studies included in this review, where physiotherapy was started on the first day post-surgery [16,17,21,22,23,25,27,28,29,30,31,32].

Higher quality standards for this scientific literature review could have been achieved, but a meta-analysis was not possible due to insufficient data on the effectiveness of autologous fat in treating temporomandibular joint (TMJ) ankylosis using interpositional arthroplasty.

This systematic review has several limitations. First, the included studies were heterogeneous in terms of study design, patient populations, follow-up periods, surgical techniques, and outcome measures, which limited the ability to perform a quantitative meta-analysis. Second, many of the studies had small sample sizes and lacked randomization, which may affect the generalizability and reliability of the findings. Additionally, the assessment of outcomes such as pain intensity and fat volume contraction was not consistently reported across all studies, making comparative analysis more challenging. Finally, although efforts were made to minimize bias through independent article screening and quality assessment, potential selection and publication bias cannot be entirely excluded.

In order to draw firm conclusions and more accurately assess the effectiveness of using different types of autologous fat in the treatment of TMJ ankylosis, more randomized clinical trials are needed, as well as additional studies investigating all three types of autologous fat: buccal fat, dermal fat, and full-thickness skin-subcutaneous fat.

## 5. Conclusions

This systematic review demonstrates that dermal fat, particularly harvested from the abdominal region, is the most commonly used and effective autologous graft material in interpositional arthroplasty for the treatment of TMJ ankylosis. It was associated with the greatest improvements in maximal mouth opening, the lowest pain intensity, and the least relative fat volume contraction when compared to other graft types. These findings suggest that dermal fat provides better functional outcomes and long-term stability in ankylosis management. However, further high-quality randomized clinical trials comparing different fat types in standardized conditions are needed to strengthen the evidence base and refine clinical guidelines.

## Figures and Tables

**Figure 1 healthcare-13-02241-f001:**
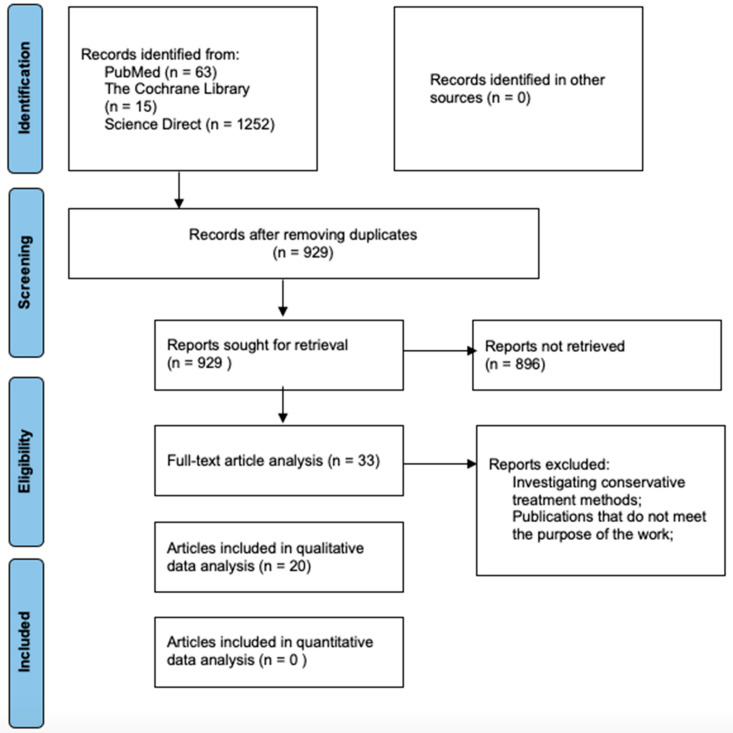
PRISMA flow diagram.

**Table 1 healthcare-13-02241-t001:** PICO Methodology.

P—pathology	Patients with temporomandibular joint ankylosis.
I—intervention	Interpositional arthroplasty with autologous fat
C—comparison	Different types of autologous fat
O—outcome	Maximum mouth opening (MMO)
Which autologous fat types are most effective in comparison to maximal mouth opening when treating TMJ ankylosis using the interpositional arthroplasty method?	

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
