# Peer review of "The Effectiveness of Using Autologous Fat in Temporomandibular Joint Ankylosis Treatment with Interposition Arthroplasty Method: A Systematic Literature Review"

_healthcare, 2025, doi:10.3390/healthcare13172241_

Round 1

Reviewer 1 Report

Comments and Suggestions for Authors

The Authors have reviewed the use of autologous fat in the treatment of TMJ ankylosis. This is an interesting subject which has been written about for some time in the OMFS literature. and has become standard in many situations. 

The main limitation of this paper is the heterogeneity of the patient populations. TMJ Ankylosis perhaps better stated as mandibular hypomobility is represented in the accepted papers as having an MMO of 15 to 0 mm. This is a big range. The causes are not defined. For example, trauma is the most common cause-were patients involved in traumatic events or surgery and how many and what previous surgeries. The treatments rendered are heterogeneous-I don't think you should compare autologous fat alone with fat + Temporalis muscle and costochondral grafts.  This confuses the picture and makes it very difficult to draw meaningful conclusions. Furthermore, the degree of boney ankylosis or classification needs to be addressed as this effects the outcome.  Reorganizing the data with groups of fat alone, fat -plus muscle, fat plus CCG/TMJ replacements might make more sense from a practical point of view.

Were all the articles available from LSMU-this is not clear.

Pain. This is very difficult to relate to the type of graft used. PO pain depends on so many risk factors and comorbidities that I would not use this as an outcome measure. Fat is used primarily to prevent recurrence so that should be the focus. 

Follow-up. 3-6 months is too short to assess the improvement in range of motion. These studies should be eliminated. 

What are the risks of fat harvesting and use? Malocclusion? At the donor site?

Author Response

Comment 1: The main limitation of this paper is the heterogeneity of the patient populations. TMJ Ankylosis perhaps better stated as mandibular hypomobility is represented in the accepted papers as having an MMO of 15 to 0 mm. This is a big range. The causes are not defined. For example, trauma is the most common cause-were patients involved in traumatic events or surgery and how many and what previous surgeries. The treatments rendered are heterogeneous-I don't think you should compare autologous fat alone with fat + Temporalis muscle and costochondral grafts.  This confuses the picture and makes it very difficult to draw meaningful conclusions. Furthermore, the degree of boney ankylosis or classification needs to be addressed as this effects the outcome.  Reorganizing the data with groups of fat alone, fat -plus muscle, fat plus CCG/TMJ replacements might make more sense from a practical point of view.

Response 1: We thank the reviewer for the insightful comments regarding the heterogeneity of the included studies and patient populations. We agree with this comment. However, one of the primary motivations for conducting this systematic review was precisely the lack of previous comprehensive reviews focused specifically on the use of autologous fat in interpositional arthroplasty for TMJ ankylosis. To our knowledge, no prior systematic review has attempted to consolidate and evaluate all potentially relevant studies with autologous fats used in interpositional arthroplasty for TMJ ankyloss - regardless of whether autologous fat was used alone or in combination with other components—into one cohesive analysis. The current literature is fragmented, and we aimed to provide an overview of all available evidence to guide future research and clinical decision-making.In many of the included studies, detailed information about the etiology of ankylosis, history of trauma or surgery, or classification/severity (e.g., Sawhney type) was not consistently reported for each patient or treatment subgroup. For this reason, we chose to systematize and present the findings as reported, and we have clearly acknowledged this heterogeneity as a limitation in both the discussion section (Lines 172-185). Our intention was not to directly compare fundamentally different treatment approaches, but rather to summarize existing outcomes where autologous fat was a component of treatment, thus providing a valuable foundation for future more targeted investigations.We hope this clarification helps explain our rationale, and we thank you again for your helpful feedback.

Comment 2: Were all the articles available from LSMU-this is not clear.

Response 2: We appreciate the reviewer’s attention to detail. As noted in section 2.3 of the Methods ("Article Search Methods"), all included full-text articles were accessible via the LSMU library, and only publications with publicly available abstracts and full texts accessible through the university system were selected. We have slightly reworded this section to make this point clearer to readers Line 153).

Comment 3: Pain. This is very difficult to relate to the type of graft used. PO pain depends on so many risk factors and comorbidities that I would not use this as an outcome measure. Fat is used primarily to prevent recurrence so that should be the focus. 

Response 3: We fully agree that postoperative pain can be influenced by numerous confounding factors, including individual pain thresholds, comorbidities, surgical technique, and perioperative management. Therefore, we acknowledge that it is not a reliable standalone outcome to assess graft effectiveness. We fully agree that postoperative pain can be influenced by numerous confounding factors, including individual pain thresholds, comorbidities, surgical technique, and perioperative management. Therefore, we acknowledge that it is not a reliable standalone outcome to assess graft effectiveness. However, pain was evaluated in several of the included studies, and was measured using standardized tools such as the Visual Analogue Scale (VAS). Since this variable was reported in the literature, we decided to summarize and present the findings in our review. Nevertheless, we have clarified in the Discussion that postoperative pain should be interpreted with caution and not considered a primary outcome (Lines 131-136).

Comment 4: Follow-up. 3-6 months is too short to assess the improvement in range of motion. These studies should be eliminated. 

Response 4: We appreciate this important observation. We agree that longer follow-up periods are preferable for evaluating the stability of improvements in range of motion and the long-term effectiveness of grafts in preventing re-ankylosis. However, we chose to include studies with shorter follow-up periods (e.g., 3–6 months) because several of these studies specifically evaluated the use of buccal fat, which is a key comparison material in our systematic review. Since buccal fat is less studied overall and data on its outcomes are relatively limited, excluding these studies would significantly reduce the ability to systematize and compare different autologous fat grafts. Moreover, even short-term outcomes can provide clinically relevant insights into early postoperative recovery, fat graft behavior, and initial joint mobility. We have clearly noted the limited follow-up duration of these studies and addressed this as a limitation in the Discussion (Lines 176-180). Our aim was to provide a comprehensive overview of all available clinical data regarding autologous fat grafting in TMJ ankylosis, and for this reason, we chose to include studies with both short- and long-term follow-up durations.

Comment 5: What are the risks of fat harvesting and use? Malocclusion? At the donor site?

Response 5: Thank you for this important question. We have addressed potential complications related to fat harvesting in the Discussion section of the manuscript. As noted, the risks at the donor site include pain, hematoma, infection, scarring, and contour irregularities, particularly when harvesting from the abdominal or iliac crest regions. Regarding intra-articular complications, although rare, improper fat placement or inadequate graft volume may affect joint biomechanics and— in some cases — contribute to malocclusion or joint instability (Lines 148-157). However, none of the included studies in our review reported severe complications directly related to fat grafting.

Reviewer 2 Report

Comments and Suggestions for Authors

The use of Autologous Fat to manage TMJ ankylosis has been a hotly debated topic in recent years, and it is useful to assess the best methodology to deliver to patients according to the available literature.

However, this systematic review presents some flaws, mainly in the methodology, that need attention, as described in my comments below.

Abstract

  • In the Materials and Methods section, information on the different steps needed for a systematic review is missing, i.e., eligibility criteria with type of articles included and keywords, and main outcome measures. Please, improve this subsection.
  • Some statements regarding the level of quality of the studies included in the review should be provided to understand if the outcomes are reliable.

Manuscript

  • This section should be improved on the rationale and impact on clinical applications. S
  • The paragraphs “2.2. Article Inclusion Criteria” and “2.4. Data Collection” may be more readable eliminating the numbered list, not useful for the scope.
  • It is not mentioned whether the search was also performed in the Grey literature or through manual search. Please, specify.
  • No information about publication bias and statistical heterogeneity is provided.
  • The selected statistical model (random / fixed effect) and the method of selection were not adequately reported.
  • There are no explanations about the method and the software used to perform statistics.
  • Some future directions and suggestions for research may be added at the end of Discussions.
Comments on the Quality of English Language

The English form can be slightly improved.

Author Response

Comment 1: Abstract. 1) In the Materials and Methods section, information on the different steps needed for a systematic review is missing, i.e., eligibility criteria with type of articles included and keywords, and main outcome measures. Please, improve this subsection. 2) Some statements regarding the level of quality of the studies included in the review should be provided to understand if the outcomes are reliable.

Response 1: Thank you for this valuable comment. We agree that clarity in the Abstract is essential for readers to quickly understand the methodology and reliability of the findings. We have revised the Materials and Methods subsection in the Abstract to explicitly include: the eligibility criteria (human clinical studies in English on TMJ ankylosis treated with autologous fat using interpositional arthroplasty), The databases searched and keywords used, and the main outcome measures, including maximal mouth opening (MMO), pain intensity, and fat volume contraction. Additionally, we have now included a statement summarizing the quality assessment of the included studies, noting that most were of moderate to high quality, as evaluated using the Cochrane RoB 2 tool and the Newcastle-Ottawa Scale. These changes have been made to enhance the transparency and methodological rigor of the Abstract, as recommended (Lines 18-29).

Comment 2: Manuscript. This section should be improved on the rationale and impact on clinical applications. The paragraphs “2.2. Article Inclusion Criteria” and “2.4. Data Collection” may be more readable eliminating the numbered list, not useful for the scope. It is not mentioned whether the search was also performed in the Grey literature or through manual search. Please, specify. No information about publication bias and statistical heterogeneity is provided. The selected statistical model (random / fixed effect) and the method of selection were not adequately reported. There are no explanations about the method and the software used to perform statistics. Some future directions and suggestions for research may be added at the end of Discussions.

Response 2: Agree. We have, accordingly modified our article to emphasize this point. In 2.2. Article Inclusion Criteria” and “2.4. Data Collection” we eliminated the numbered list. Also we added information about grey literature and manual searches (Lines 140-146), information about publication bias and statistical heterogeneity, selected statistical model (Lines 203-2010), used software (Lines 168-170) and suggestetions for research in Discussion (Lines 186-190).

Reviewer 3 Report

Comments and Suggestions for Authors

Abstract

  1. The abbreviation MOM has already been explained, yet in line 25 it is wrongly mentioned as maximum MOM. This is confusing. Please revise.
  2. In the results mention the different types of autologous fat grafts reported in the different studies.
  3. Conclusion – Is there any particular anatomic region from which superior quality dermis graft is usable for interpositional arthroplasty?

Introduction

  1. Mention in brief the etiology and pathogenesis of TMJ ankylosis.
  2. Other than fat grafts, are there any other grafts or substitute materials used for gap arthroplasty? Mention about them in brief and justify why fat grafts are superior.
  3. There is no justification explaining the need for this review. Include that in the introduction.
  4. Aim and objectives mean the same. Revise them to be mentioned as Primary and secondary objectives of the study.

Methods

  1. In the section about search methods, include the degree of internal consistency between the two reviewers.

Results

  1. Why is the study data presented in two tables (4 and 5)? Kindly explain the difference in data presented in these two tables. If no differences, it would be prudent to merge these two tables.
  2. In several places, the results are mentioned as multiple short sentences. Please follow a paragraph structure.

Discussion and conclusion

  1. Include a paragraph about the limitations of the study.
  2. Revise the conclusion to a paragraph structure.
  3. Since most of the conclusion points towards favorable outcomes for derma fat graft, include a statement about why and how derma fat graft could be a better clinical alternative for gap arthroplasty.
  4. Combine the practical recommendations as part of the conclusion.
Comments on the Quality of English Language

The overall quality of English grammar and sentence structuring needs improvement with the help of a native speaker.

Author Response

Comment 1: the abbreviation MOM has already been explained, yet in line 25 it is wrongly mentioned as maximum MOM. This is confusing. Please revise.

Response 1: Thank you for pointing this out. We agree and we removed word maximum (Line 13).

Comment 2: In the results mention the different types of autologous fat grafts reported in the different studies.

Response 2: Thank you for your comment. Agree. We modified our article and mentioned the different types of autologous fat grafts reported in different studies (Line 12-14)

Comment 3: Conclusion – Is there any particular anatomic region from which superior quality dermis graft is usable for interpositional arthroplasty?

Response 3: Agree. We modified our Discussion and included this information in Lines 138-147.

Comment 4: Mention in brief the etiology and pathogenesis of TMJ ankylosis.

Response 4: Agree. We added etiology and pathogenesis of TMJ ankylosis (Lines 52-58).

Comment 5: Other than fat grafts, are there any other grafts or substitute materials used for gap arthroplasty? Mention about them in brief and justify why fat grafts are superior.

Response 5: Agree. We added this information in Introduction Lines 61-65.

Comment 6: There is no justification explaining the need for this review. Include that in the introduction.

Response 6: Agree. We explained justification in Introduction Lines 95-100.

Comment 7: Aim and objectives mean the same. Revise them to be mentioned as Primary and secondary objectives of the study.

Response 7: Agree. We change it to Primary and secondary objectives (Line 103-113).

Comment 8: In the section about search methods, include the degree of internal consistency between the two reviewers.

Response 8: Agree. We provided this information in 2.3 paragraph (Lines 160-162).

Comment 9: Why is the study data presented in two tables (4 and 5)? Kindly explain the difference in data presented in these two tables. If no differences, it would be prudent to merge these two tables.

Response 9: Agree. Thank you for this observation. In response to your suggestion, we have now merged the data from Tables 4 and 5 into a single comprehensive table (Table 4) to avoid redundancy and improve clarity. This unified table includes all relevant study characteristics and outcome data in one place for easier interpretation.

Comment 10: In several places, the results are mentioned as multiple short sentences. Please follow a paragraph structure.

Response 10: Agree. We corrected results paragraph structure (Lines 38-78).

Comment 11: Include a paragraph about the limitations of the study.

Response 11: Agree. We included paragraph about the limitations of the study (Lines 176-191).

Comment 12: Revise the conclusion to a paragraph structure.

Response 12: Agree. We removed number list and revised conclusion structure (Lines 193-201)

Comment 13: Since most of the conclusion points towards favorable outcomes for derma fat graft, include a statement about why and how derma fat graft could be a better clinical alternative for gap arthroplasty.

Response 13: We appreciate the reviewer’s observation regarding the favorable outcomes associated with dermal fat grafts. While our findings do suggest that dermal fat—particularly from the abdominal region—may be associated with better postoperative results (e.g., maximal mouth opening, lower pain intensity, and reduced fat contraction), we have also emphasized in the Discussion that the included studies are clinically and methodologically heterogeneous, with variability in surgical techniques, patient populations, and follow-up durations. For this reason, we have intentionally avoided making definitive clinical recommendations in the conclusion. Instead, we state that while dermal fat shows promise, more high-quality, standardized, and comparative studies are needed to confirm its superiority over other graft materials in gap arthroplasty. We believe this cautious interpretation aligns with the current strength of available evidence and upholds the standards of systematic review reporting (Lines 138-146).

Comment 14: Combine the practical recommendations as part of the conclusion.

Reponse 14: Thank you for your recommendations. Based on your comment, we reviewed the Conclusion and ensured that the key findings were clearly summarized.

Reviewer 4 Report

Comments and Suggestions for Authors

Based on a meta-analysis of 20 publications involving data from 369 patients, the authors draw conclusions related to temporomandibular joint arthroplasty. It has been established that the most commonly used types of autologous fat in interpositional arthroplasty for ankylosis are dermal fat.

I consider the "Practical Recommendations" chapter important, as it provides practical summary advice for surgeons performing arthroplasty:

  • It is advisable to select dermal fat due to better maximum mouth opening, lower pain intensity, and early initiation of physiotherapy.

Furthermore, it is important to note that the relatively fat volume contraction after surgical treatment of temporomandibular joint ankylosis with interpositional arthroplasty was achieved using dermal fat.

In summary: I find the paper good and interesting, and I recommend it for publication. However, I have two comments that it would be good for the authors to consider:

  1. In the Introduction, I suggest briefly listing the surgical options available for TMI ankylosis.
  2. What is the usual surgical approach used for performing the arthroplasty?

After addressing these two brief questions, I recommend the work for publication.

Author Response

Comment 1: In the Introduction, I suggest briefly listing the surgical options available for TMI ankylosis.

Response 1: Thank you for the suggestion. As recommended, we have included a brief overview of the main surgical options for TMJ ankylosis in the Introduction section, including gap arthroplasty, interpositional arthroplasty, total joint reconstruction, and condylectomy with distraction osteogenesis or costochondral grafting (Lines 61-63).

Comment 2: What is the usual surgical approach used for performing the arthroplasty?

Response 2: Agree. We mentioned additional information about surgery approach (Lines 65-74)

Reviewer 5 Report

Comments and Suggestions for Authors

Dear Authors,

Thank You for a pleasure to read Your work.

I have several comments to improve Your article.

Sincerely, Reviewer

Abstract

Lines 14-15: fat is not very popular method and exactly using word ‘popular’ is not right for scientific paper. Often using method is endoprothesis from different materials but it is also depending on the type of ankylosis.

Please, change these lines.

Please, check the key words according to MeSH.

Introduction

Lines 46-47: the management depends on type of ankylosis. If You tell about surgical treatment  then, please, write about bony ankylosis.

Materials and Methods

Figure 1: please, make it better quality as the resolution is rather bad.

Subsection 2.2

If You tell about systematic review it must include ONLY CONTROL RANDOMIZED TRIALS for high evidence statement. If Your review analyses other types of study then it is not classic systematic review and it would have lower significant for medicine.

If You have no enough articles for systematic review You could add additional

section with analysis of other types of study but it would not make Your article the classic systematic review. Thus, the inclusion criteria are discussable.

The same are for meta-analysis and other systematic review. You could use their results for introduction and discussion but they could not take a part in the main text of the article.

Subsection 2.3

Please, write more clear search strategy with different words combinations.

Subsection 3.1

Please, replace figure 1 to this section for better understanding.

Subsections 3.2-3.3

Tables 2-5. Please, do not mix the types of study, separate them at least for two tables for RCT and other types of study with subsections (cohort, non randomized etc) for better understanding.

Discussion

Lines 131-136. Please, write study limitations in subsection and spread it.

Acknowledgment

It is better to write ‘we’ as You have 3 authors.

Author Response

Comment 1: Lines 14-15: fat is not very popular method and exactly using word ‘popular’ is not right for scientific paper. Often using method is endoprothesis from different materials but it is also depending on the type of ankylosis. Please, change these lines.

Response 1: Thank you for pointing this out. We change these lines (Lines 13-16).

Comment 2: Please, check the key words according to MeSH.

Response 2: Agree. We modified key words according to MeSH (Lines 150-153).

Comment 3: Lines 46-47: the management depends on type of ankylosis. If You tell about surgical treatment  then, please, write about bony ankylosis.

Response 3: Agree. We have revised the text to clarify that surgical treatment is primarily indicated for bony ankylosis, while fibrous ankylosis may respond to conservative measures. This distinction has now been made explicit in the revised version of the manuscript (Lines 43-48).

Comment 4: Figure 1: please, make it better quality as the resolution is rather bad.

Response 4: Thank you for pointing this out. We have now replaced Figure 1 with a higher-resolution version to ensure better clarity and readability for the final publication.

Comment 5: Subsection 2.2 If You tell about systematic review it must include ONLY CONTROL RANDOMIZED TRIALS for high evidence statement. If Your review analyses other types of study then it is not classic systematic review and it would have lower significant for medicine. If You have no enough articles for systematic review You could add additional section with analysis of other types of study but it would not make Your article the classic systematic review. Thus, the inclusion criteria are discussable. The same are for meta-analysis and other systematic review. You could use their results for introduction and discussion but they could not take a part in the main text of the article.

Response 5: We thank the reviewer for this important comment and fully acknowledge the distinction between systematic reviews based solely on randomized controlled trials (RCTs) and those that include other study designs. In our case, only a small number of RCTs were available on the specific topic of autologous fat use in interpositional arthroplasty for TMJ ankylosis. Given the limited number of high-level studies, and in order to provide a more comprehensive overview of the existing evidence, we decided to include both randomized and non-randomized clinical studies (e.g., prospective and retrospective cohort studies) that met strict inclusion criteria and focused on relevant clinical outcomes. We have clearly stated this in the Methods (Section 2.2), and we assessed the risk of bias using appropriate tools: the Cochrane RoB 2 tool for RCTs and the Newcastle-Ottawa Scale for observational studies. Moreover, we have addressed the heterogeneity and lower level of evidence as a limitation in the Discussion section. Our intent to systematize all available clinical evidence on this under-researched topic to guide future, more robust studies.

Comment 6: Subsection 2.3 Please, write more clear search strategy with different words combinations.

Response 6: Agree. We modified our 2.3 section according to your comment (Lines 141-164).

Comment 7: Tables 2-5. Please, do not mix the types of study, separate them at least for two tables for RCT and other types of study with subsections (cohort, non randomized etc) for better understanding.

Response 7: Thank you for your valuable suggestion. In response, we have merged the previous two tables into a single comprehensive table and have clearly organized the data into separate subsections for randomized controlled trials (RCTs) and other study types (prospective and retrospective cohort studies). This restructuring improves clarity and allows for better comparison of study designs and quality. I will attach the table separately.

Comment 8: Lines 131-136. Please, write study limitations in subsection and spread it.

Response 8: Agree. We added limitations of the study in Lines 172-191.

Comment 9: It is better to write ‘we’ as You have 3 authors.

Response 9: Agree. We changed that in Acknowledgement section.

Round 2

Reviewer 1 Report

Comments and Suggestions for Authors

The Authors have adequately addressed the Reviewers comments and have clarified the limitations of the study 

Author Response

Comment: The Authors have adequately addressed the Reviewers comments and have clarified the limitations of the study 

Response: On behalf of all the authors, I would like to thank you and the reviewers for your constructive feedback and valuable suggestions. We appreciate the time and effort dedicated to improving our manuscript.

Reviewer 5 Report

Comments and Suggestions for Authors

Dear Authors,

Thank You for Your corrections.

Please, add for table 4 the separate column with probabilities from study as I previously asked You.

Sincerely, Reviewer

Author Response

Comment 1: Please, add for table 4 the separate column with probabilities from study as I previously asked You.

Response 1: 

We thank the reviewer for the suggestion to include a separate column with p-values in Table 4. Unfortunately, in most of the included studies, p-values were either not reported or were presented only for comparisons between different surgical interventions (e.g., dermal fat vs. temporalis muscle), rather than for the fat graft group alone. In some cases, the results were pooled for combined interventions, making it impossible to extract a p-value specific to autologous fat without introducing bias or misinterpretation.

Because of this heterogeneity in reporting and the lack of consistent statistical data across studies, we believe that adding a p-value column to Table 4 could be misleading. Instead, we have retained the original format of the table and have clearly described these limitations in the Discussion section (Line 254-258), emphasizing that the statistical significance of the results should be interpreted with caution.